# Impact of COVID-19 Diagnosis on Mortality in Patients with ST-Elevation Myocardial Infarction Hospitalized during the National Outbreak in Italy

**DOI:** 10.3390/jcm11247350

**Published:** 2022-12-10

**Authors:** Leonardo De Luca, Stefano Rosato, Paola D’Errigo, Barbara Giordani, Gian Francesco Mureddu, Gabriella Badoni, Fulvia Seccareccia, Giovanni Baglio

**Affiliations:** 1Department of Cardio-Thoracic and Vascular Medicine and Surgery, Division of Cardiology, A.O. San Camillo-Forlanini, 00152 Rome, Italy; 2Faculty of Medicine and Dentistry, UniCamillus—Saint Camillus International University of Health Sciences, 00131 Rome, Italy; 3National Centre for Global Health, Istituto Superiore di Sanità, 00161 Rome, Italy; 4Italian National Agency for Regional Healthcare Services, 00161 Rome, Italy; 5Division of Cardiology, San Giovanni Addolorata Hospital, 00184 Rome, Italy

**Keywords:** acute myocardial infarction, percutaneous coronary intervention, national outbreak, COVID-19

## Abstract

Background. We sought to assess the clinical impact of COVID-19 infection on mortality in patients with ST-elevation myocardial infarction (STEMI) admitted during the national outbreak in Italy. Methods. We analysed a nationwide, comprehensive, and universal administrative database of consecutive STEMI patients admitted during lockdown for COVID-19 infection (11 March–3 May 2020) and the equivalent periods of the previous 5 years in Italy. The observed rate of 30-day and 6-month all-cause mortality of STEMI patients with and without COVID-19 infection during the lockdown was compared with the expected rate of death, according to the trend of the previous 5 years. Results. During the study period, 32.910 STEMI hospitalizations occurred in Italy. Among these, 4048 STEMI patients were admitted during the 2020 outbreak: 170 (4.2%) with and 3878 (95.8%) without a COVID-19 diagnosis. According to the 5-year trend, the 2020 expected rates of 30-day and 6-month all-cause mortality were 9.2% and 12.6%, while the observed incidences of death were 10.8% (*p* = 0.016) and 14.4% (*p* = 0.017), respectively. Excluding STEMI patients with a COVID-19 diagnosis, the mortality rate resulted in accordance with the prior 5-year trend. After multiple corrections, the presence of COVID-19 diagnosis was an independent predictor of all-cause mortality at 30 days [adjusted odds ratio (OR) 4.5; 95% confidence intervals (CI) 3.09–6.45; *p* < 0.0001] and 6 months (adjusted OR 3.6; 95% CI: 2.47–5.12; *p* < 0.0001). Conclusions.During the 2020 national outbreak in Italy, COVID-19 infection significantly increased the mortality trend in patients with STEMI.

## 1. Introduction

The pandemic of coronavirus disease-2019 (COVID-19) has caused an unprecedented disaster in terms of deaths worldwide. Italy, which has about 59 million inhabitants, was the first European nation to be affected by COVID-19 and was caught unprepared, with around 18.5 million confirmed total cases and more than 168,000 deaths to date [1,2]. The pandemic has mainly affected the north of Italy, where, especially in the first half of 2020, most confirmed cases of COVID-19 and related fatal events occurred [1,2].

Italy has been the first western country to enact anationwide lockdownfor controlling the pandemic of coronavirus disease-2019 (COVID-19), implementing measures that were considered the most radical against the outbreak in Europe [1,2,3]. The government of Italy imposed a nationalcontainment, officially starting from 11 March and ending, after two additional decrees prolonging the national outbreak, on 3 May 2020, restricting the movement of the population except for necessity, work and health circumstances [4,5]. The national lockdown was the period with the widest spread of COVID-19 and the highest rate of mortality related to the infection, during which only patients with health emergencies were admitted to hospitals in Italy [6].

We aimed to analyse the clinical impact of COVID-19 infection on mortality in patients with ST-elevation myocardial infarction (STEMI), admitted during the national outbreak, and in the equivalent periods of the previous 5 years in Italy using a nationwide, comprehensive and universal administrative database.

## 2. Methods

### 2.1. Study Design

This was a retrospective cohort study that enrolled consecutive patients admitted to all public and private hospitals in Italy for a STEMI event during the national outbreak ofCOVID-19 (11 March–3 May 2020) and the equivalent periods of the 5 previous years. We compared baseline characteristics, hospitalization rates and 30-day and 6-month all-cause mortality between STEMI patients admitted during the national outbreak for COVID-19 in 2020 and the prior 5-year equivalent periods. The Italian National Registry of Hospital Discharge Records (HDR), provided by the Italian Ministry of Health (MoH), and other administrative databases available through a collaboration with the Italian National Program for Outcome Evaluation (PNE-AGENAS) were used as sources of data.

### 2.2. Study Population

All patients in the HDR, aged 18 to 100 years, resident in Italy, admitted during the study period and reporting a primary diagnosis of STEMI (International Classification of Disease, 9th Revision, Clinical Modification [ICD 9 CM]: 410.1–410.6 and 410.8) or a secondary diagnosis of STEMI, with any concomitant complication within the primary diagnosis (ICD-9-CM codes 411, 413, 414, 426, 427, 428, 423.0, 429.5, 429.6, 429.71, 429.79, 429.81, 518.4, 518.81, 780.01, 780.2, 785.51, 799.1, 997.02 and 998.2), were selected [7].

Patients discharged home within 2 days from admission (probable false STEMI cases) were excluded. Furthermore, to avoid the inclusion of multiple admissions due to the same event, duplicate records, and records concerning both transfers of patients to another hospital and patients with a previous AMI admission within 30 days from the index admission were excluded [8].

According to the Italian MoH official documents for COVID-19 case identification released between March and October 2020, STEMI patients with a concomitant definite or suspected diagnosis of COVID-19 were defined as STEMI cases, with at least one of the following ICD 9 CM codes: 078.89 Other specified diseases due to viruses (MoH first guidelines—20 March 2020); 043COVID-19 disease, 480.4 COVID-19 Pneumonia, 518.9 COVID-19 Acute respiratory distress syndrome (ARDS), 519.7 COVID-19 Other respiratory infections (MoH Decree—28 October 2020); 079.82 SARS-associated coronavirus, 480.3 Pneumonia due to SARS-associated coronavirus (ICD-9-CM codes for SARS); codes identifying exposure, isolation, anamnesis, observation (‘V01.85’, ‘V01.79’, ‘V71.83’, ’V07.0’, ‘V71.84’, ‘V07.00’, ‘V12.04’, ‘V01.82’) or pneumonia in other infectious diseases (484.8) [9].

Data on patient risk factors and comorbidities, according to the ICD9-CM codes reported in Appendix A, were retrieved either from the index admission or the previous 5-year hospitalizations. The rate of percutaneous coronary intervention (PCI), performed within the first 2 days of hospital admission, was also assessed.

In order to evaluate the impact of COVID-19 infection on STEMI mortality in different areas of the country, Italy was divided into three macro-regions: Northern (Lombardia, Piemonte, Valle d’Aosta, Veneto, Friuli Venezia-Giulia, Trentino Alto-Adige, Liguria and Emilia-Romagna; accounting for a total of 13,480,648 inhabitants in 2020), Central (Lazio, Toscana, Umbria and Marche; 5,719,084 inhabitants) and Southern (Abruzzo, Molise, Puglia, Basilicata, Campania, Calabria, Sicilia and Sardegna; 9,850,364 inhabitants).

The 30-day and 6-month all-cause mortality represented the main adverse outcomes.

### 2.3. Statistical Analysis

Prevalence of risk factors and comorbidities were presented as counts and percentages; age was expressed as the mean ± standard deviation.

The number of expected STEMI events and the rates of the comorbidities and outcomes in the 2020 national outbreak were estimated by a linear regression model using the number of STEMI events, and the rates of the comorbidities and outcomes in the prior 5-year equivalent periods as predictors. The numbers of the actual and expected events in the 2020 study period were compared by the Poisson test. The observed and expected rates of both comorbidities and outcomes were compared using the log-normal distribution property of the rate ratio (H0: observed rate/expected rate = 1).

The normal distribution of continuous parameters was tested with the Kolmogorov–Smirnov test. Variables with a skewed distribution were compared with the use of Wilcoxon rank sum tests. *t*-Test, Chi-square or Fisher exact tests were used to compare frequencies among COVID-19 and non-COVID-19 patients in the 2020 STEMI cohort, as appropriate.

To provide adjusted outcome data, age, gender and patients’ risk factors and comorbidities were included in the multivariate models as potential confounding factors; stepwise logistic procedures were used to identify independent associations with each of the considered outcomes. Since some chronic comorbidities recorded in the index hospitalization show a paradoxical protective effect [8], the same comorbidities recorded in the previous hospitalizations were also forced into the models.

All assumptions of statistical methods were explicitly checked. Statistical analyses were performed using SAS 9.4 (Cary, NC, USA).

## 3. Results

During the study period, 32.910 STEMI hospitalizations occurred in Italy. In the almost 8 weeks of the 2020 national outbreak, 4048 STEMI patients were admitted at 365 centres in Italy: 170 (4.2%) with and 3878 (95.8%) without a COVID-19 diagnosis. Patients with a COVID-19 infection were older and more frequently had a history of cerebrovascular diseases and anaemia, compared to STEMI patients without COVID-19 (Appendix A).

Compared to the previous 5 years, the rate of admissions for STEMI during the lockdown in 2020 was markedly reduced (from 6085 in 2015 to 4048 in 2020; percentage change −33.5). Considering the 5-year trend, the observed number of STEMI admissions in 2020 was significantly reduced as compared to the expected rate (4048 vs. 5523; *p* < 0.0001) (Figure 1A). The incidences of STEMI developed during hospitalization were 13.0% and 15.9%, for those with and without a concomitant COVID-19 diagnosis, respectively. The reduced rate in STEMI admissions during the lockdown, as compared to the expected rate based on the previous 5-year trend, was consistent in Northern, Central and Southern Italy (all *p* values <0.0001) (Figure 1B). Although the absolute number of STEMI admissions decreased, the relative percentage of STEMI patients receiving a PCI increased in the study period, as compared to the previous 5 years of observation (from 69.5% in 2015 to 79.9% in 2020; percentage change +15.0%). This trend was consistent in North (from 68.3% in 2015 to 77.0% in 2020; percentage change +12.7%), Central (from 71.7% in 2015 to 81.7% in 2020; percentage change +13.9) and South (from 70.0% in 2015 to 83.1% in 2020; percentage change +18.7%) Italy (Appendix A).

Demographic and clinical characteristics of STEMI patients admitted during the 2020 lockdown were comparable with those admitted during the equivalent periods in the previous 5 years of observation, except for the history of PCI that was significantly reduced in the 2020 population (from 14.5% in 2015 to 12.6% in 2020; *p* < 0.0001) (Table 1).

### Mortality Trends

According to the 5-year trend, the 2020 expected rate of 30-day all-cause mortality was 9.2%, while the observed incidence of death was 10.8% (*p* = 0.016). Excluding STEMI patients with a COVID-19 diagnosis, the observed incidence of 30-day mortality was 9.9% (*p* = 0.33 compared to the expected trend rate) (Figure 2). Accordingly, the 2020 expected rate of 6-month all-cause mortality was 12.6%, while the observed incidence of death was 14.4% (*p* = 0.017); after excluding STEMI patients with COVID-19, the observed incidence of mortality at 6 months was 13.5% (*p* = 0.25 compared to the expected trend rate) (Figure 3). The difference in the observed rates of 30-day and 6-month mortality among STEMI patients admitted during the 2020 lockdown, with and without COVID-19 infection, was particularly evident in Northern Italy (Figure 2 and Figure 3).

After multiple corrections, the presence of COVID-19 diagnosis was an independent predictor of all-cause mortality at 30 days [adjusted odds ratio (OR) 4.5; 95% confidence intervals (CI) 3.09–6.45; *p* < 0.0001] (Appendix A) and 6 months (adjusted OR 3.6; 95% CI: 2.47–5.12; *p* < 0.0001) (Table 2).

## 4. Discussion

In this retrospective analysis of nationwide administrative data, we observed a marked reduction in hospital admissions for STEMI during the national outbreak for COVID-19 in Italy, with a higher-than-expected all-cause death at both 1 and 6 months, compared with the mortality trend for STEMI in the same calendar period in the previous 5 years. Notably, when STEMI patients with a concomitant COVID-19 diagnosis were excluded from the analysis, the observed mortality rate resulted in accordance with the prior 5-year trend.

Several studies have documented marked reductions in prevalence, significant changes in management and poorer outcomes of STEMI during the COVID-19 pandemic [10,11,12,13,14,15]. A retrospective analysis of patients admitted for acute coronary syndromes at 15 hospitals in northern Italy during the early days of the COVID-19 pandemic, documented a significant decrease in hospitalization rates and an inexplicable increase in mortality [16]. In a large retrospective cohort study of STEMI patients admitted between 2019 and 2020 at 509 centres in the United States, a concomitant diagnosis of COVID-19 was significantly associated with higher rates of in-hospital mortality [17]. Notably, the clinical database used in this latest study mainly included academic medical centres, substantially limiting the generalisation of the obtained results [17]. In addition, a full year of theCOVID-19 pandemic was used for observation, with different restriction measures that varied across states in North America. In our study, we firstly analysed a nationwide dataset of STEMI patients with a concomitant COVID-19 diagnosis admitted to Italian hospitals during the period of maximum social constrain and the highest mortality rate related to the pandemic (almost 1000 daily deaths in Italy) [6]. We also compared the study cohort and the death rate at 1 and 6 months, with the trend of the 5 years prior to the national outbreak for COVID-19. This allowed us to reliably compare the observed andexpected death rates based on mortality trends.

Patient concerns about a referral to emergency departments, due to the fear of a possible in-hospital infection, was suggested as a critical reason for the decline in STEMI admissions [13,14]. In this regard, the reasons for the increased mortality in STEMI patients with a concomitant COVID-19 diagnosis observed in our study may be attributed to longer times to revascularization, due to the need for additional precautions related to infection in centres not predisposed to receive COVID-19 patients; infection-related conditions due to the severity of pulmonary involvement (especially in the first pandemic phase included in our observation period); or to prothrombotic complications that have further worsened or triggered the STEMI context. Accordingly, several reports suggested that STEMI might be caused by extensive microvascular thrombosis in the absence of epicardial coronary obstruction or might be complicated by a higher-than-anticipated rate of stent thrombosis during COVID-19 infection [18,19,20,21]. Other observations reported significant delays in the management of STEMI patients during the pandemic [22]. In this regard, a retrospective registry performed in European high-volume PCI centres suggested that the pandemic was associated with a significant increase in door-to-balloon and total ischaemia times. These findings may explain data from North American or European registries enrolling patients with confirmed or suspected COVID-19 undergoing PCI for STEMI who demonstrated a significantly higher in-hospital mortality during the pandemic, compared to historical control periods [14,23,24]. In our series there was no reduction in the expected number of PCI for STEMI during the lockdown, as compared to the previous 5-year trend, even though the absolute number of STEMI admissions was reduced. This suggests that during the national outbreak, PCI for STEMI was still performed and was not perceived as futile, even in patients with concomitant COVID-19 infection; although, it is possible that in-hospital management, including time to revascularization, was suboptimal in those patients.

### Limitations

There are several limitations to using an administrative health claims database. One is that the lack of specific clinical information may have affected the accuracy of the diagnosis, severity, and risk stratification of STEMI.Indeed, some prognostic data, such as vital signs, instrumental parameters, time to reperfusion and procedural detail were not available. In addition, we could only assess the number of PCI performed within 2 days of hospital admission for STEMI, since the time of admission was not available on the HDR. Another limitation is the deficiency in the ICD-9 CM code descriptions to provide comprehensive data on in-hospital complications and cause of death. Finally, we cannot determine the extent to which misclassification and coding errors may be present.

## 5. Conclusions

The number of STEMI admissions during the 2020 national outbreak in Italy was markedly reduced, while the rate of PCI was consistent with the 5-year expected trend. A concomitant diagnosis of COVID-19 infection significantly increased the mortality rate; indeed, after excluding STEMI patients with a diagnosis of COVID-19, the rate of observed mortality was in trend with the previous 5 years.

Further studies are warranted to understand the mechanisms underlying the association between COVID-19 and mortality in the STEMI context.

## Figures and Tables

**Figure 1 jcm-11-07350-f001:**
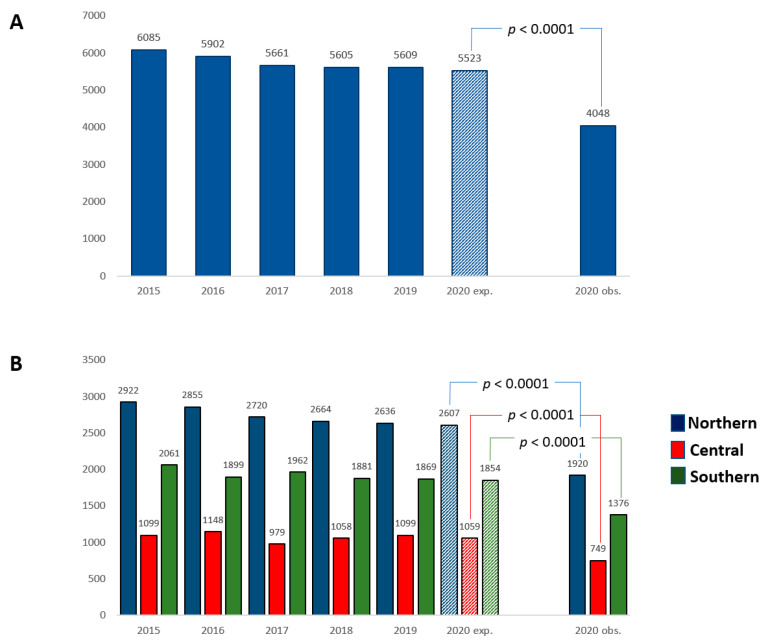
Expected (exp.) and observed (obs.) incidence of STEMI admission during the 2020 national outbreak and over the equivalent periods in the previous 5 years in Italy (**A**) and by geographic regions (**B**).

**Figure 2 jcm-11-07350-f002:**
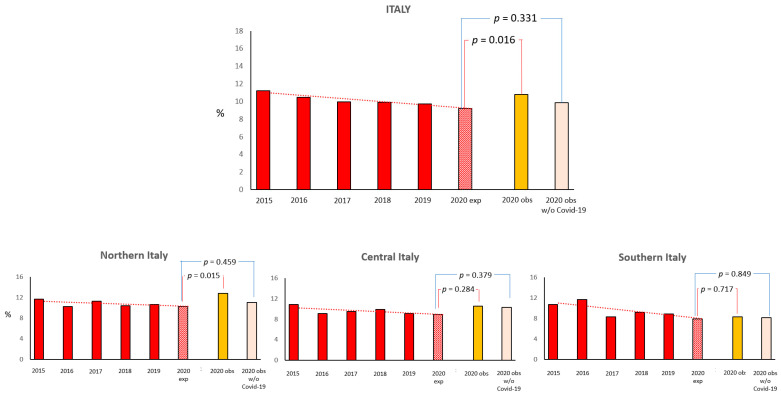
Expected (exp.) and observed (obs.) mortality rate at 30 days (in the overall STEMI population and excluding those with COVID-19 infection) during the 2020 national outbreak and over the equivalent periods in the previous 5 years in Italy and by geographic regions.

**Figure 3 jcm-11-07350-f003:**
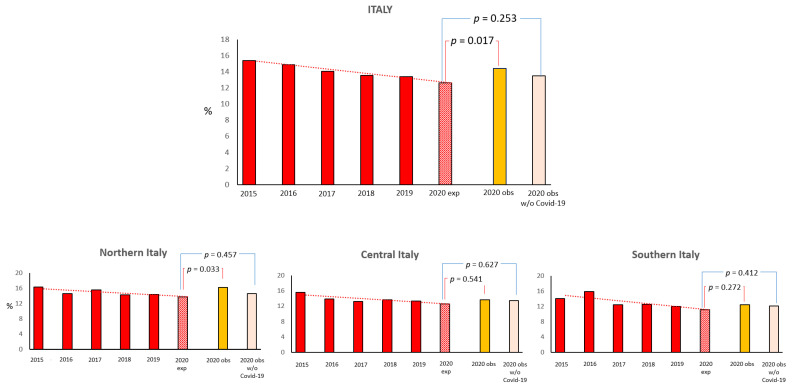
Expected (exp.) and observed (obs.) mortality rate at 6 months (in the overall STEMI population and excluding those with COVID-19 infection) during the 2020 national outbreak and over the equivalent periods in the previous 5 years in Italy and by geographic regions.

**Table 1 jcm-11-07350-t001:** Baseline characteristics of the enlisted population by year.

	2015 (*n* = 6085)	2016 (*n* = 5902)	2017 (*n* = 5661)	2018 (*n* = 5605)	2019 (*n* = 5609)	2020 (*n* = 4048)	*p* Value *
Age (years), mean ± SD	67.5 ± 13.6	67.8 ± 13.6	67.5 ± 13.3	67.2 ± 13.6	66.8 ± 13.0	67.9 ± 13.9	0.147
Gender (females), *n* (%)	1866 (30.7)	1747 (29.6)	1596 (28.2)	1592 (28.4)	1559 (27.8)	1028 (25.4)	0.136
Hypertension, *n* (%)	959 (15.8)	895 (15.2)	791 (14.0)	748 (13.3)	676 (12.1)	475 (11.7)	0.530
Diabetes, *n* (%)	542 (8.9)	509 (8.6)	445 (7.9)	421 (7.5)	425 (7.6)	265 (6.5)	0.454
Heart failure, *n* (%)	302 (5)	281 (4.8)	257 (4.5)	258 (4.6)	227 (4.0)	144 (3.6)	0.312
Ill-defined descriptions and complications of heart disease, *n* (%)	65 (1.1)	81 (1.4)	49 (0.9)	47 (0.8)	52 (0.9)	30 (0.7)	0.883
Cerebrovascular disease, *n* (%)	442 (7.3)	367 (6.2)	323 (5.7)	319 (5.7)	274 (4.9)	173 (4.3)	0.836
Cerebrovascular disease (ind. adm.), *n* (%)	206 (3.4)	181 (3.1)	168 (3.0)	158 (2.8)	140 (2.5)	86 (2.1)	0.514
Vascular disease, *n* (%)	251 (4.1)	252 (4.3)	208 (3.7)	197 (3.5)	185 (3.3)	134 (3.3)	0.511
Vascular disease (ind. adm.), *n* (%)	160 (2.6)	127 (2.2)	124 (2.2)	124 (2.2)	103 (1.8)	79 (2.0)	0.493
Chronic coronary syndromes, *n* (%)	587 (9.6)	549 (9.3)	498 (8.8)	498 (8.9)	436 (7.8)	315 (7.8)	0.800
Arrhythmias, *n* (%)	366 (6.0)	301 (5.1)	319 (5.6)	278 (5.0)	263 (4.7)	196 (4.8)	0.393
Anaemia, *n* (%)	192 (3.2)	185 (3.1)	170 (3.0)	138 (2.5)	153 (2.7)	117 (2.9)	0.207
Anaemia (ind. adm.), *n* (%)	156 (2.6)	142 (2.4)	159 (2.8)	153 (2.7)	143 (2.5)	93 (2.3)	0.246
Blood clotting defects, *n* (%)	5 (0.1)	6 (0.1)	8 (0.1)	8 (0.1)	8 (0.1)	2 (0.0)	0.122
Blood clotting defects (ind. adm.), *n* (%)	1 (0.0)	4 (0.1)	1 (0.0)	1 (0.0)	2 (0.0)	1 (0.0)	0.935
Other hematological diseases, *n* (%)	45 (0.7)	32 (0.5)	19 (0.3)	28 (0.5)	29 (0.5)	17 (0.4)	0.779
Other hematological diseases (ind. adm.), *n* (%)	23 (0.4)	23 (0.4)	19 (0.3)	33 (0.6)	31 (0.6)	22 (0.5)	0.678
Cardiomyopathy, *n* (%)	53 (0.9)	50 (0.8)	33 (0.6)	31 (0.6)	41 (0.7)	15 (0.4)	0.248
Cardiomyopathy (ind. adm.), *n* (%)	81 (1.3)	80 (1.4)	68 (1.2)	63 (1.1)	75 (1.3)	60 (1.5)	0.278
Rheumatic heart disease, *n* (%)	48 (0.8)	35 (0.6)	25 (0.4)	33 (0.6)	22 (0.4)	14 (0.3)	0.850
Rheumatic heart disease (ind. adm.), *n* (%)	41 (0.7)	40 (0.7)	41 (0.7)	41 (0.7)	50 (0.9)	22 (0.5)	0.070
Endocarditis and acute myocarditis, *n* (%)	4 (0.1)	4 (0.1)	4 (0.1)	9 (0.2)	8 (0.1)	3 (0.1)	0.210
Other chronic heart conditions, *n* (%)	49 (0.8)	67 (1.1)	41 (0.7)	45 (0.8)	44 (0.8)	28 (0.7)	0.804
Other chronic heart conditions (ind.adm.), *n* (%)	72 (1.2)	86 (1.5)	80 (1.4)	81 (1.4)	72 (1.3)	43 (1.1)	0.155
Chronic kidney disease, *n* (%)	245 (4)	247 (4.2)	205 (3.6)	209 (3.7)	209 (3.7)	139 (3.4)	0.793
Chronic kidney diseases (ind. adm.), *n* (%)	431 (7.1)	408 (6.9)	356 (6.3)	350 (6.2)	331 (5.9)	248 (6.1)	0.292
Other chronic disease (liver, pancreas, intestine), *n* (%)	85 (1.4)	95 (1.6)	76 (1.3)	81 (1.4)	62 (1.1)	40 (1.0)	0.465
Other chronic disease (liver, pancreas, intestine) (ind. adm.), *n* (%)	25 (0.4)	34 (0.6)	28 (0.5)	24 (0.4)	18 (0.3)	11 (0.3)	0.539
Obesity, *n* (%)	78 (1.3)	80 (1.4)	73 (1.3)	56 (1.0)	56 (1.0)	49 (1.2)	0.185
Obesity (ind. adm.), *n* (%)	149 (2.4)	145 (2.5)	140 (2.5)	138 (2.5)	156 (2.8)	100 (2.5)	0.471
Chronic obstructive pulmonary disease, *n* (%)	278 (4.6)	229 (3.9)	228 (4)	182 (3.2)	185 (3.3)	104 (2.6)	0.433
Malignant neoplasms, *n* (%)	449 (7.4)	390 (6.6)	409 (7.2)	434 (7.7)	378 (6.7)	273 (6.7)	0.533
Previous MI	654 (10.7)	575 (9.7)	530 (9.4)	529 (9.4)	497 (8.9)	341 (8.4)	0.978
Previous vascular surgery, *n* (%)	223 (3.7)	227 (3.8)	173 (3.1)	200 (3.6)	185 (3.3)	138 (3.4)	0.570
Previous cerebral revascularization, *n* (%)	59 (1)	38 (0.6)	38 (0.7)	43 (0.8)	36 (0.6)	18 (0.4)	0.398
Previous CABG	174 (2.9)	169 (2.9)	132 (2.3)	133 (2.4)	128 (2.3)	82 (2.0)	0.943
Other previous cardiac surgery than CABG, *n* (%)	58 (1.0)	51 (0.9)	59 (1.0)	54 (1.0)	61 (1.1)	43 (1.1)	0.895
Previous PCI	883 (14.5)	860 (14.6)	796 (14.1)	831 (14.8)	879 (15.7)	510 (12.6)	<0.0001

* The *p* values refer to the comparison between the observed and expected rates of comorbidities in the 2020 study period; Abbreviations: CABG: coronary artery bypass grafting; MI: myocardial infarction; PCI: percutaneous coronary intervention; Note: ind. adm—comorbidity information retrieved at the index admission.

**Table 2 jcm-11-07350-t002:** Logistic regression model for 6 months mortality.

	Crude OR	Adjusted OR	CI 95%	*p* Value
Gender (female)	2.6	1.2	1.117–1.299	<0.0001
Age (years)	1.1	1.1	1.072–1.079	<0.0001
COVID-19	3.4	3.6	2.465–5.122	<0.0001
Chronic coronary syndromes	2.3	1.3	1.148–1.475	<0.0001
Previous vascular surgery	3.1	1.8	1.531–2.080	<0.0001
Other chronic heart conditions	3.5	1.6	1.201–2.158	0.001
Anaemia	4.6	1.4	1.200–1.651	<0.0001
Cardiomyopathy	3.0	1.4	1.003–1.952	0.048
Diabetes	2.7	1.3	1.128–1.422	<0.0001
Heart failure	5.0	1.5	1.305–1.727	<0.0001
Cerebrovascular disease	3.3	1.2	1.101–1.399	<0.0001
Other chronic disease (liver, pancreas, intestine) (ind. adm.)	1.7	1.5	0.958–2.428	0.075
Chronic kidney diseases (ind. adm.)	2.9	1.2	1.041–1.321	0.009
Obesity (ind. adm.)	0.4	0.6	0.467–0.862	0.004
Previous PCI	0.6	0.6	0.501–0.638	<0.0001
Malignancy	2.6	1.7	1.536–1.911	<0.0001
Previous CABG	0.9	0.6	0.449–0.711	<0.0001
PCI ≤ 2 days	0.2	0.4	0.409–0.474	<0.0001

Abbreviations: CABG: coronary artery bypass grafting; PCI: percutaneous coronary intervention; Note: ind. adm—comorbidity information retrieved at the index admission.

## Data Availability

The data underlying this article were provided by AGENAS/ISS by permission. Data will be shared on request to the corresponding author with the permission of AGENAS/ISS.

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
