# Peer review of "Impact of COVID-19 Diagnosis on Mortality in Patients with ST-Elevation Myocardial Infarction Hospitalized during the National Outbreak in Italy"

_jcm, 2022, doi:10.3390/jcm11247350_

Round 1
Reviewer 1 Report
1- The first two paragraphs of the introduction are not related to the research topic. I will suggest cutting it down and adding more information about the research gap/question this paper is going to answer
2- Do you think that the reason behind the decreased number of STEMI patients during the pandemic is that patients feared seeking medical care due to fear of infection? You may want to expand on that in the discussion.
3- Do you have information on the cause of death of patients with COVID-19? Was it related to the STEMI or non-related (e.g., pneumonia, respiratory failure, etc.)?
4- Did you adjust for the modality of treatment received for these patients? Essentially, the 30-day mortality of patients getting PCI is going to be different from those who did not.
5- For patients with COVID-19, how many patients developed STEMI while hospitalized for COVID-19? This may shed some light on the high risk of thromboembolism in COVID-19 patients, especially if they did not have risk factors for MI.
Author Response
- The first two paragraphs of the introduction are not related to the research topic. I will suggest cutting it down and adding more information about the research gap/question this paper is going to answer
We thank the reviewer for the suggestion. We rephrased the initial part of the introduction, as suggested.
- Do you think that the reason behind the decreased number of STEMI patients during the pandemic is that patients feared seeking medical care due to fear of infection? You may want to expand on that in the discussion.
We have now added in the discussion (page 9, last paragraph) the following sentence ‘Patient concerns about a referral to emergency departments due to the fear of a possible in-hospital infection was suggested as a critical reason for the decline in STEMI admissions’.
- Do you have information on the cause of death of patients with COVID-19? Was it related to the STEMI or non-related (e.g., pneumonia, respiratory failure, etc.)?
Unfortunately, we are unable to have information on the cause of death of patients with COVID-19. We derive the information on the vital status from the nationwide Tax Registry.
- Did you adjust for the modality of treatment received for these patients? Essentially, the 30-day mortality of patients getting PCI is going to be different from those who did not.
We thank the reviewer for the comment. We have now included the PCI in the model and corrected the OR and CI for all variables in Table 2 and Suppl Table 3 and, for Covid-19, also in the abstract and text. The main results did not change.
- For patients with COVID-19, how many patients developed STEMI while hospitalized for COVID-19? This may shed some light on the high risk of thromboembolism in COVID-19 patients, especially if they did not have risk factors for MI.
We have now added on page 7, second paragraph the following sentence ‘The incidence of STEMI developed during the hospitalization was 13.0% and 15.9%, for those with and without a concomitant Codiv-19 diagnosis, respectively.’.
Reviewer 2 Report
“The difference in the observed rates of 30-day and 6-momth mortality among STEMI”- should be “month”
“After multiple corrections, the presence of Covid-19 diagnosis resulted one of the
independent predictors of all-cause mortality at 30 days”
The above sentence is one of the crucial ones, please exchange a word “resulted” to another one or just re-arrange a sentence to underline the problem.
I have had some trouble with reading tables and finding them as they are not properly named.
Sometimes authors use (Suppl. Table 3) and sometimes (Figure S1) and SUPPL FIG. 1, which means it needs to be unified. Probably it should be mentioned in the text that there is also a supplement. Then, you shoud put the proper numbers /names of such figures an
Author Response
“The difference in the observed rates of 30-day and 6-momth mortality among STEMI”- should be “month”
We have now corrected the word in the results section (page 8, first paragraph).
“After multiple corrections, the presence of Covid-19 diagnosis resulted one of the independent predictors of all-cause mortality at 30 days”. The above sentence is one of the crucial ones, please exchange a word “resulted” to another one or just re-arrange a sentence to underline the problem.
We rephrased the sentence in both abstract and text.
I have had some trouble with reading tables and finding them as they are not properly named. Sometimes authors use (Suppl. Table 3) and sometimes (Figure S1) and SUPPL FIG. 1, which means it needs to be unified. Probably it should be mentioned in the text that there is also a supplement. Then, you shoud put the proper numbers /names of such figures and tables.
We have now uniformed the name of supplementary materials